# Effect of Meal and Whole Larvae of Black Soldier Fly (*Hermetia illucens*) on the Performance, Blood Lipid Profile, Slaughter Characteristics, Sensory Properties and Fatty Acid Composition of Pheasant (*Phasianus colchicus* L.) Muscles

**DOI:** 10.3390/ani15213215

**Published:** 2025-11-05

**Authors:** Grzegorz Rytlewski, Marian Flis, Eugeniusz R. Grela

**Affiliations:** 1Polish Hunting Association Gdańsk District Board, 80-286 Gdańsk, Poland; goigr@wp.pl; 2Department of Animal Ethology and Wildlife Management, Animal Sciences and Bioeconomy, University of Life Sciences in Lublin, 20-950 Lublin, Poland; 3Institute of Animal Nutrition and Bromatology, Animal Sciences and Bioeconomy, University of Life Sciences in Lublin, 20-950 Lublin, Poland; eugeniusz.grela@up.lublin.pl

**Keywords:** pheasant, meal and larvae of black soldier fly, performance, blood, carcass traits, fatty acids

## Abstract

The composition and nutritional value of the pheasant diet can significantly impact egg yield, chick survival rate, and the nutritional and dietary value of eggs and meat. The present study determined the effectiveness of replacing 50% or 100% of soybean meal with black soldier fly (BSF) products in the form of meal or whole dried larvae on the performance and slaughter characteristics, blood lipid profile, sensory properties, and fatty acid composition of the muscles of the pheasant (*Phasianus colchicus* L.), considering gender. The study showed that BSF products can successfully replace between 50% and 100% of soybean meal.

## 1. Introduction

The first introductions of pheasants to Central Europe occurred between 500 and 800 AD, involving Caucasian pheasants (*Phasianus colchicus*). Other pheasant species, including *Phasianus colchicus mongolicus* and *Phasianus colchicus torquatus*, were introduced to Europe considerably later [1]. Pheasants were recorded in the Silesia region of Poland in the 16th century as a mix of several subspecies. Initially, they were mainly regarded as park animals, but gradually became common throughout the country. Successful acclimatisation combined with broad adaptive capabilities resulted in the rapid growth of their population, and pheasants thus became game animals [2,3]. In recent years, there has been a decline in the small game population, which has also affected pheasants. The main reasons cited for this decline include predation, disease, and low survival rates among chicks in the first weeks of life, attributed to the advancing changes in their natural habitat. Hence, attempts are being made in many regions to introduce these birds, based on material obtained from breeding farms [4,5,6,7,8]. Although breeding activities in such cases focus on obtaining material for introduction with characteristics similar to those of wild birds, the effectiveness of such measures is relatively low. Therefore, newer solutions are continually being sought to maintain and optimise diets, aiming to produce the best material for introduction that is suited to the natural conditions of habitats [2,8,9,10,11,12].

Due to their valuable features, especially the nutritional and dietary benefits of pheasant meat, pheasants are increasingly kept for fattening, in addition to confined breeding for introduction purposes [13,14,15]. Even though pheasant farming for high-quality meat is a relatively new phenomenon, it is becoming increasingly common. With its high protein content and low fat content, combined with a favourable fatty acid profile, particularly the PUFA n-3 to n-6 ratio, pheasant meat is a highly nutritious food whose value exceeds that of broiler chicken meat. It is also recognised as a dietary and health-promoting raw material [13,14,15,16,17,18,19,20,21,22].

In recent years, improved nutritional solutions have been sought to optimise reproduction processes, bird growth rates, and the quality of meat and its subsequent suitability for consumption or storage [14,23,24,25,26,27]. To this end, various attempts are being made at optimising the levels of protein, energy, vitamins and minerals in pheasant feed, as these are the major factors determining production results [13,19,28,29,30]. Moreover, efforts have been made to optimise nutrition by incorporating insect meal or larvae into bird diets [31,32,33,34]. Research in this field has shown that supplementing the diet with animal protein, instead of plant protein, has a favourable effect on production in terms of egg yield and chick rearing, as well as immune function and improvement of intestinal microflora [35,36,37,38]. Moreover, studies conducted on pheasants have shown that replacing soybean protein with insect meal has a positive effect on the number of eggs laid, chick survival, and weight gain, while also reducing the weight of breast muscles. At the same time, a slight decrease in the level of monounsaturated fatty acids (MUFA) in eggs was noted [8,11].

The aim of the study was to determine the effect of replacing 50% or 100% of soybean meal with black soldier fly (*Hermetia illucens*) products in the form of meal or whole dried larvae on performance and slaughter characteristics, blood lipid profile, sensory properties, and fatty acid composition of the muscles of the pheasant (*Phasianus colchicus* L.), considering gender.

## 2. Materials and Methods

### 2.1. Birds and Housing

The study was conducted on an experimental farm situated in the village of Stara Kiszewa, in the Pomerania region of Poland (18°10′ E, 53°59′ N). The study material consisted of pheasants (*Phasianus colchicus*), reared for four weeks on a standard feed mixture (Table 1), and then kept in groups under a confined system in aviaries with an area of 72 m^2^ for each group (control group and four experimental groups). The aviaries were four metres high and made of flexible netting, allowing the birds to fly around without injuring themselves while preventing them from escaping. Each aviary was equipped with two nipple drinkers along with a feeder (80 cm long, i.e., 4.0 cm of feeder edge per bird). Each aviary housed 10 hens and 10 cocks. After the rearing period, all pheasants, at five weeks of age, were randomly assigned to individual groups and fed *ad libitum* a granular mixture with the components and chemical compositions provided in Table 1. The nutritional value of the diet was similar to that recommended by the National Research Council [39].

### 2.2. Diet

The experiment involved 5-week-old pheasants that were fed, in separate nutritional groups, a Grower & Finisher feed mixture containing varying levels of insect meal and dried insect larvae. The control group diet included plant-based feed ingredients commonly used in pheasant nutrition (Table 1). In experimental groups, insect meal (group II—50% HM and III—100% HM) and whole dried insect larvae (groups IV—50% HL and V—100% HL) were used instead of soybean meal with a proportion of GMOs. The insect meal and dried insect larvae were supplied by HiProMine S.A. from Robakowo, Poland. Black soldier fly (BSF) larvae were fed a mixture of plant by-products [40,41]. To obtain full-fat insect meal, the insect biomass was frozen at −20 °C and then dried at 50 °C for 24 h. It was subsequently homogenised using a hammer mill to ensure the diameter of the resulting particles was less than 0.1 mm. The raw material was hygienised by heating it to 100 °C for 95 min, in accordance with the recommendations of Annex IV of Commission Regulation No 141/2011 laying down health rules as regards animal by-products. Mixtures of experimental feed with the addition of insect products were prepared using dried insect larvae or meal produced from them, with 50% or 100% of soybean meal replaced by these insect products. In each case, all feed ingredients were mixed and granulated at 60 °C.

During the experiment, the birds were weighed at 5, 8, 12 and 16 weeks of age. Daily feed intake was monitored by weighing the mixture into feeders and noting any uneaten feed. At the end of the experiment, all 16-week-old pheasants were weighed, blood was collected from the wing vein, and the birds were decapitated. The birds were slaughtered at a poultry slaughterhouse located 35 km from the breeding farm, in accordance with procedure 1099/2009, under veterinary supervision [42]. Twelve hours before blood collection, the animals had no access to their diet. Blood was collected from the wing vein just before slaughter into 2 mL heparin tubes under veterinary supervision. Three muscle groups were dissected free: breast, shank and thigh muscles. Internal organs, including the heart, muscular stomach, liver and head, were collected and weighed on an AXIS ATZ 2200 laboratory scale (Axis, Gdańsk, Poland). The dissected free muscles were stored in a freezer at −20 °C until chemical analysis.

### 2.3. Analytical Procedures

The chemical composition (dry matter, crude protein, crude fibre, crude fat, crude ash and fatty acids) of the control and experimental diets was analysed according to AOAC procedures [43]. Test kits developed by Cormay (Lublin, Poland) were used to determine the content of triglycerides (TG), total cholesterol (TCH), and high-density lipoprotein cholesterol (HDL). Low-density lipoprotein cholesterol (LDL) was calculated using the formula given by Friedewald et al. [44]: LDL (mmol L^−1^) = total cholesterol − HDL − triglycerides/2.2

The total cholesterol content in the muscles was determined using the chromatographic method proposed by Botsoglou et al. [45]. Cholesterol was identified by comparing the sample retention times with those for the authenticated laboratory standard. Quantification was performed in relation to an external standard based on a graphically represented curve with cholesterol levels and peak area values. The cholesterol concentrations for the individual muscle groups were determined in mg/100 g.

The fatty acid profile of extracted muscle lipids [46] and the ether extract of feed was determined by a gas chromatographic method using a Varian 450-GC chromatograph (Varian, Inc., Palo Alto, CA, USA). The control, collection and calculation of results were performed using Galaxie™ Chromatography Data System version 1.9.3.2 software. The technical conditions of the chromatographic method for the separation of fatty acids were as follows: capillary column: Select™ Biodiesel for FAME (60 m, 0.32 mm, 0.25 μm, Windsor, CT, USA); stationary phase: Select Biodiesel for FAME Fused Silica; Column oven: initial temperature of 100 °C, and final temperature of 240 °C; detector: FID temperature of 270 °C; carrier gas: helium, flow rate of 1.5 mL/min. The sensory properties of meat (aroma, succulence and tenderness) were assessed according to the scale provided by Baryłko-Piekielna [47]. A 5-point scale was used to assess the aroma and intensity, with the following points assigned for aroma: 5—very distinct, 4—distinct, 3—slightly distinct, 2—perceptible, 1—imperceptible. In turn, when assessing the succulence, the point ranges indicated the following: 5—succulent, 4—moderately succulent, 3—slightly succulent, 2—slightly dry, 1—dry. The tenderness scale included the following scores: 5—very tender, 4—tender, 3—slightly tender, 2—firm, and 1—very firm. The samples for evaluation were cooked in a 0.6% table salt solution with a water-to-meat ratio of 2:1 until the meat reached an internal temperature of 72 °C. After heat treatment, the samples were cooled, cut into equal pieces, and evaluated by persons trained in sensory evaluation [48]. The sensory quality of the meat was assessed by a team with significant experience in the sensory evaluation of meat and meat products (3–8 years of experience). The study was conducted in a room with a temperature of 22 ± 1 °C, during daylight hours. Each assessor was provided with hot, sugar-free tea to neutralise the taste between each sample assessment.

### 2.4. Statistical Analysis

Statistical analysis was conducted using the STATISTICA 13.3 program (TIBCO Software, Inc., Palo Alto, CA, USA). Analyses of normal distribution and homogeneity of variance were performed using the Shapiro–Wilk test and Bartlett’s test. To determine the significance of differences between groups, a one-way analysis of variance was performed, and the significance level was assessed using Tukey’s test, at *p* < 0.05.

## 3. Results

The contents of basic nutrients and metabolic energy in the experimental and control mixtures were similar and appropriate for the chick rearing period (Table 2). The fatty acid profile showed a significant increase in the proportion of lauric acid (C12:0) and myristic acid (C14:0), and a decrease in the proportion of oleic acid (18:1, n-9) in mixtures containing meal or whole *Hermetia illucens* insect larvae, especially when they substituted 100% of the soybean meal content.

The body weight of birds randomly assigned to individual groups was similar (193 ± 4 g), with significant differences noted between genders: males were approximately 24 g heavier than females (Table 3). After 16 weeks of fattening, slight differences in body weight were observed among the groups of birds, resulting in variations in daily weight gains. The most effective growth was noted for pheasants fed a mixture containing *Hermetia illucens* insect larvae meal in groups 50 HM and 100 HM. Significant differences were observed in the daily weight gains, with males gaining 11.8 g compared to 9.5 g for females. Daily feed intake ranged from 56.1 g in the 50 HM group to 57.9 g in the control group, with males consuming 4 g more feed than females (Table 3).

Substituting soybean meal with insect products had no significant effect on the weight of individual muscles, the head, and internal organs, including the liver, heart or stomach (Table 4). Significant differences in the analysed indicators were found for the gender, with cocks showing higher values compared to hens. No significant interactions were noted between nutrition and the gender of growing pheasants.

The serum lipid profile and cholesterol levels in the breast and thigh muscles were not significantly dependent on the feeding regimen or the gender of the pheasants (Table 5).

An analysis of fatty acid content in the breast muscles showed differences between individual feeding groups and bird gender, both for breast muscles (Table 6) and thigh muscles (Table 7). Significant differences were found between the feeding groups and the control group in the breast muscle fat content of C18:3, n-3, and C20:0 acids. Significant differences were also observed in the content of C18:2, n-6 acid, but only between the genders of the birds. These results affected the overall content of PUFA, n-3 acids, with the highest proportion noted in the HM100 and HL100 feeding groups. In contrast, the PUFA, n-6 acid group showed differences only between the genders of the birds. The PUFA n-6/n-3 acid ratio proved to be significant between groups, with the lowest value noted for the HL100 group, which differed statistically significantly from the other feeding groups. Hens exhibited a significantly lower value of this indicator in the breast muscles, as compared to cocks. For thigh muscles, differences in the contents of individual fatty acids were also noted between the feeding groups and the control group. Additionally, a statistically significant difference in fatty acid content was observed between the leg muscles and the breast muscles. The total content of SFAs and MUFAs was higher in the thigh muscles, while the total content of PUFA, n-3 and PUFA, n-6 acids was higher in the breast muscles. The PUFA n6/n-3 acid ratio was significantly higher in the breast muscles, with this value being statistically significant compared to the thigh muscles.

Analysis of the sensory qualities of the breast and thigh muscles revealed no significant differences in the assessment of aroma, smell and fibrousness, depending on the nutritional factors applied (Figure 1, Figure 2, Figure 3 and Figure 4). Despite the lack of significant differences, in the case of cocks, the highest values of the analysed characteristics were recorded for the 50 HM feeding group for the breast muscles and the 50 HL feeding group for the thigh muscles. Regarding the breast muscles in females, the highest score for the analysed parameters was observed in the 50 HM feeding group, whereas the lowest score was noted in the 100 HL group. In the thigh muscle group, the highest score was given to the 100 HM and 50 HL feeding groups, whereas the lowest score, as for the breast muscles, was given to the 100 HL feeding group. In all the feeding groups, the muscles of hens showed higher sensory qualities. The characteristics that were predominant in the sensory evaluation included taste in hens and cocks, as well as aroma in cocks.

## 4. Discussion

The use of insect meal in the feeding of growing pheasants represents an innovative direction in research on alternative protein sources in animal production. The results of previous experiments suggest that replacing traditional protein components, such as soybean meal or fish meal, with insect products (e.g., from *Hermetia illucens* larvae) can yield tangible benefits in both sustainable animal production and the utilisation of waste products in the diet of insects. Insect production is also characterised by lower greenhouse gas emissions, lower water consumption and fewer agricultural by-products compared to traditional sources of plant and animal proteins. Insect-based products can serve as a significant component in sustainable animal nutrition strategies, which is also becoming increasingly important in poultry farming and breeding, including game and ornamental species [11,40,50,51].

For confined pheasant breeding, one of the major factors determining the optimisation of rearing and breeding is their nutrition. In recent years, alternative sources of dietary supplementation for birds, including pheasants, that regulate protein and fat content and quality have been sought [13,19,28,52]. Increasing interest is being focused on the use of insect products, including meals from whole or de-fatted larvae [48]. Black soldier fly (BSF) larvae are increasingly recognised as a balanced and nutritious component of feed for various animals, including poultry. Additions of BSF products to monogastric animal feed improve growth, meat quality and metabolic functions, as well as the overall welfare of animals [53]. Many studies have shown that the inclusion of BSF larvae (live, full-fat dried or partially de-fatted) in poultry diets has a beneficial effect on growth, performance, nutrient digestibility, pathogen resistance and gut microbiota [15,31,54]. A study conducted on laying hens showed a positive effect of such additives on the caecal microbiome, and no effect on apprehension [55]. In contrast, a study conducted in Italy showed that supplementing hens’ diets with dried or live insect larvae had no adverse effect on meat quality and improved the fatty acid profile in muscles, which is beneficial for consumers [56].

Research into the effect of adding insect larvae meal or whole insects has also been conducted on farms breeding wild game birds, including pheasants. The use of BSF meal in the diet of these birds contributed to an increased proportion of muscles in the carcasses, accompanied by lower feed consumption. The experimental groups in which BSF was used were characterised by higher protein and fat contents and a lower water content. No significant changes were noted in the amino acid profile or the mineral composition of the muscles [57]. Another study showed that adding BSF to the diet of laying pheasant hens increased egg yield while reducing the proportion of defective eggs. The dietary supplementation applied had a positive effect on the chemical composition of the eggs [11]. In turn, a study conducted by Rytlewski et al. [34] on the effect of pheasant hens’ egg yield showed that the best nutritional option appeared to be one containing whole insect larvae, which replaced 100% of soybean meal in feed mixtures. Supplementation of soybean meal with *Hermetia illucens* contributed to an increase in the proportion of saturated fatty acids (SFAs) and the n-6/n-3 acid ratio, without a significant effect on AI, TI or h/H in pheasant eggs. Supplementing the diet with insect meal diversifies it and makes its composition more similar to that of the food found in the natural habitat of game birds. A study involving a diet containing insect meal, conducted on pheasant chicks in England, showed that after release into the natural environment, they performed better in searching for and digesting natural food, thanks to the development of proper intestinal morphology [58]. In contrast, in Japanese quails (*Coturnix japonica*), adding insect meal to their feed increased the feed conversion ratio while reducing carcass and breast weight, with no change in the fatty acid composition [59]. Another study on quail egg yield found that the addition of insect meal affected both egg yield and egg quality, while increasing the content of saturated fatty acids [60]. A study conducted by Suparman et al. [61] on Japanese quails showed that different levels of dietary supplementation with BSF meal had a significant effect on egg production and shell thickness.

In turn, a study conducted on broiler chickens [62,63] revealed that partially replacing soybean meal with insect meal improved the food conversion ratio and the birds’ body weight, which may have analogous applications in pheasant breeding. Similar observations were noted in a study on grey partridges [64], in which the addition of *Hermetia illucens* larvae meal promoted weight gain. A study on Barbary partridge meat (*Alectoris barbara*), in which a varied addition of *Tenebrio molitor* (TM) and *Hermetia illucens* (HI) meal was used as a source of protein in their diet, reported an increase in the weight of individual muscles, without any evident impact on the quality and sensory properties of individual muscles [65]. Another study on Barbary partridges demonstrated that the addition of *Hermetia illucens* meal to the birds’ diet significantly affected the count and diversity of bacteria in the caecal microbiome, thereby improving intestinal health and stabilising the microbial environment [66]. In contrast, a study conducted on hens showed that the addition of BSF had an effect on both the quality of eggs and the blood fat content in laying hens [67]. In another study on laying hens, the addition of BSF contributed to an increase in the total lipid, phospholipid and triglyceride contents in blood serum. The same study also noted changes in the lipid composition of egg yolk, specifically an increase in total lipid and triglyceride levels. There was also an improvement in the egg yield and hatching rate [68].

In the presented research, the content of protein, energy, and other nutrients in the experimental mixtures was similar to that of the control group’s mixtures, depending on the rearing period. This similarity allowed for the assessment of the influence of the experimental factor used in the nutrition of growing pheasants. This allowed the impact of the experimental factor applied in the feeding of growing pheasants to be assessed. The chemical composition and nutritional value of BSF-containing products used in the present study were previously reported [34] and were similar to those found in other studies [33,50,69]. Significant differences in experimental mixtures for groups receiving a proportion of meal (HM50 and HM100) or whole larvae (HL50 and HL100) were found in the contents of lauric (C12:0) and myristic (C14:0) acids, which was a consequence of their high content in BSF products. The high content of C12:0 acid in the mixtures significantly increased its proportion in the breast and thigh muscle fat. A study on Barbary partridges (*Alectoris barbara*) that replaced 50% of the plant protein with *Hermetia illucens* resulted in a significant increase in C12:0 and C16:1n-7 contents, as well as an increase in AI and TI indices, without affecting cholesterol levels [68]. The significant pressure of lauric acid originating from BSF in the fatty acid composition of animal products was also confirmed by a study conducted by Rytlewski et al. [34] on the lipid profile in pheasant eggs. A study on the effect of dietary supplementation on meat quality through various additions of BSF in Japanese quails revealed that there was a change in the fatty acid profile, especially in the breast muscles. There was an increase in the total proportion of SFAs, especially the C12:0 and C14:0 acids [70]. These acids, particularly lauric acid, can have a supportive effect on the birds’ health due to their antibacterial properties. This was also confirmed by Spranghers et al. [71], who found that lauric acid restricted the growth of intestinal pathogens in piglets. In the case of growing pheasants, this may improve the survivability of chicks and the condition of the flock in both confined and free-range rearing.

Substituting soybean meal with BSF products in the form of meal (HM) or whole larvae (HL) did not produce significant changes in the birds’ slaughter weight, daily weight gain, and feed intake. Furthermore, the weight of the analysed muscles and internal organs, i.e., the heart, liver, and stomach, as well as the blood lipid profile and cholesterol content in both analysed muscles, were not significantly dependent on the substitution of soybean meal with BSF products. Moreover, the sensory evaluation of thigh and breast muscles did not deteriorate after the use of BSF products in amounts substituting 50% or 100% of soybean meal in mixtures for growing pheasants. It can therefore be concluded that BSF-containing products can be fully utilised in pheasant nutrition. What is also noteworthy is the favourable fatty acid profile in the breast, where an increased proportion of n-3 PUFAs and a reduced n-6/n-3 PUFA ratio were noted for both muscles.

## 5. Conclusions

Insect meal or whole dried larvae can become a valuable component in feeding growing pheasants, improving rearing results. Insect meal has a high protein content with a favourable amino acid profile [31,33], similar to traditional animal-derived raw materials or soybean meal. The results obtained from the study showed that, regardless of the type of BSF products (e.g., meal or whole dried larvae), they can be successfully used to replace soybean meal, which is most commonly available commercially in the form containing GMOs. Improvements in daily body weight gain were achieved in pheasants, especially those receiving insect larvae meal in their diet, without significant changes in musculature, heart, liver, and stomach weight, blood lipid profile, and muscle cholesterol. The muscle fat of pheasants receiving insect larvae (*Hermetia illucens*) products in their diet was characterised by a significantly higher content of lauric acid (C12:0) and n-3 PUFA, and a favourable n-6/n-3 PUFA ratio for consumers. Further research is required, however, to optimise the proportion of BSF products in feed rations and to determine their impact on the quality parameters of meat, eggs, and feathers, which is particularly important in the case of species intended for introduction for hunting purposes, as well as those intended for direct consumption of meat and eggs from birds kept in aviaries.

## Figures and Tables

**Figure 1 animals-15-03215-f001:**
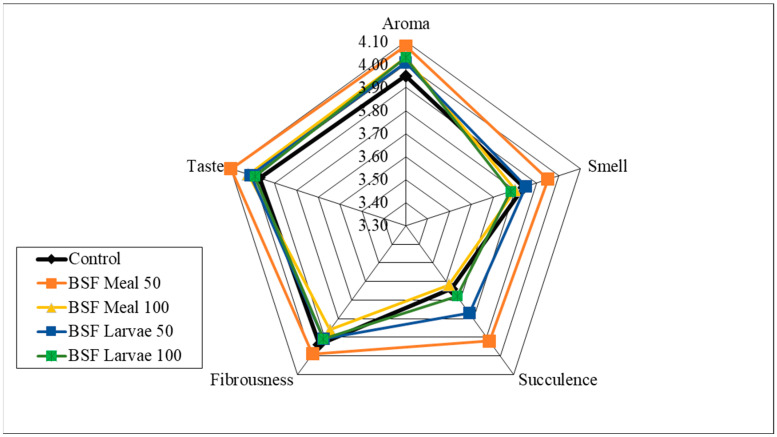
Sensory characteristics of the breast muscle of male pheasants.

**Figure 2 animals-15-03215-f002:**
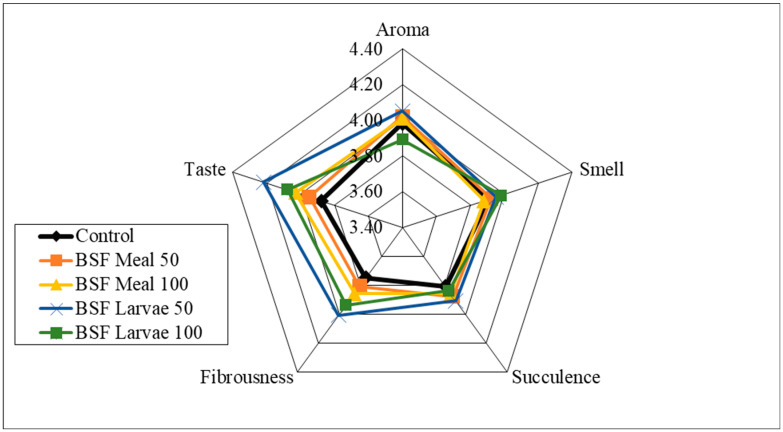
Sensory characteristics of the thigh muscle of male pheasants.

**Figure 3 animals-15-03215-f003:**
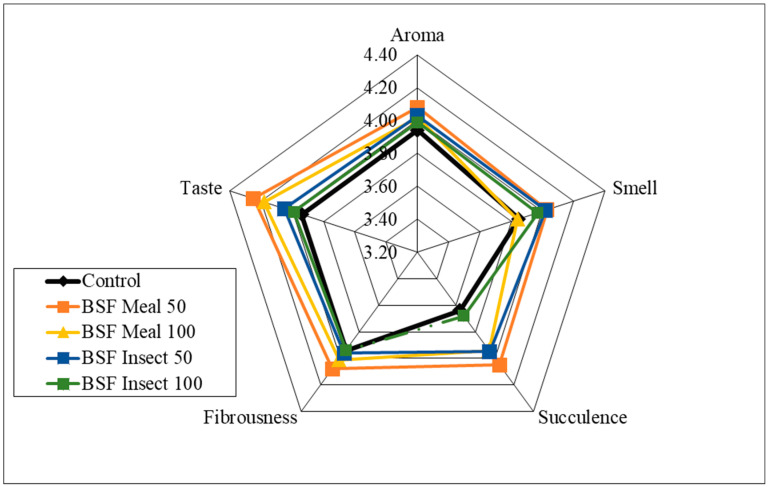
Sensory characteristics of the breast muscle of female pheasants.

**Figure 4 animals-15-03215-f004:**
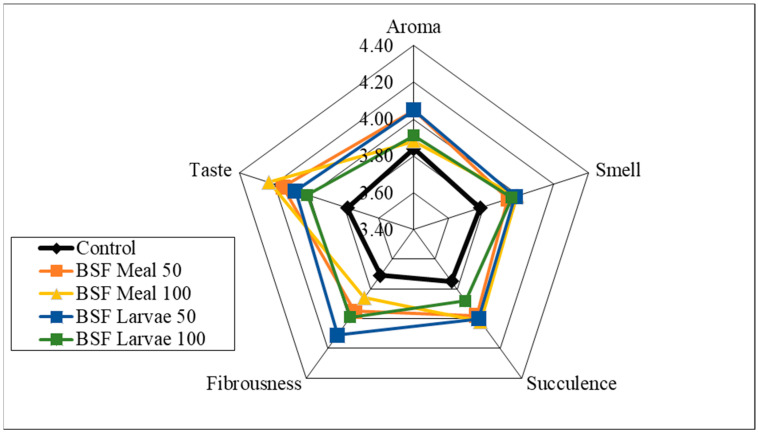
Sensory characteristics of the thigh muscle of female pheasants.

**Table 1 animals-15-03215-t001:** Ingredients (g kg^−1^) of reared pheasant diet with insect larvae meal or whole dried larvae.

Item	Reared (0–4 Weeks)	Grower (5–8 Weeks) Period					Finisher (9–16 Weeks) Period				
		Control	50 HM	100 HM	50 HL	100 HL	Control	50 HM	100 HM	50 HL	100 HL
Corn	232.1	272.3	297.3	322.3	297.3	322.3	241.6	266.6	291.6	266.6	291.6
Wheat	100.0	100.0	100.0	100.0	100.0	100.0	160.0	160.0	160.0	160.0	160.0
Soybean meal	280.0	280.0	140.0	0.0	140.0	0.0	250.0	125.0	0.0	125.0	0.0
Garden pea	50.0	50.0	50.0	50.0	50.0	50.0	50.0	50.0	50.0	50.0	50.0
Fish meal	80.0	20.0	20.0	20.0	20.0	20.0	0.0	0.0	0.0	0.0	0.0
Insect meal	0.0	0.0	140.0	280.0	0.0	0.0	0.0	125.0	250.0	0.0	0.0
Whole dried larvae	0.0	0.0	0.0	0.0	140.0	280.0	0.0	0.0	0.0	125.0	250.0
Linseed	40.0	40.0	40.0	40.0	40.0	40.0	40.0	40.0	40.0	40.0	40.0
Sunflower meal	80.0	80.0	80.0	80.0	80.0	80.0	80.0	80.0	80.0	80.0	80.0
Soybean oil	50.0	50.0	25.0	0.0	25.0	0.0	50.0	25.0	0.0	25.0	0.0
Sorghum	10.0	30.0	30.0	30.0	30.0	30.0	50.0	50.0	50.0	50.0	50.0
Dicalcium phosphate	16.0	16.0	16.0	16.0	16.0	16.0	17.0	17.0	17.0	17.0	17.0
Calcium carbonate	55.0	55.0	55.0	55.0	55.0	55.0	55.0	55.0	55.0	55.0	55.0
Salt	3.0	3.0	3.0	3.0	3.0	3.0	3.0	3.0	3.0	3.0	3.0
Mineral-vitamin premix	2.5	2.5	2.5	2.5	2.5	2.5	2.5	2.5	2.5	2.5	2.5
DL-methionine	0.5	0.4	0.4	0.4	0.4	0.4	0.3	0.3	0.3	0.3	0.3
L-lysine chloride	0.9	0.8	0.8	0.8	0.8	0.8	0.6	0.6	0.6	0.6	0.6

The mineral-vitamin premix in the control group provided in a 1 kg diet: Mn 60 mg, I 1 mg, Fe 54 mg, Zn 100 mg, Cu 11 mg, Se 0.2 mg, vit. A 10,000 IU, vit. D_3_ 2500 IU, vit. E 50 mg, vit. K_3_ 2 mg, vit. B_1_ 1.5 mg, vit. B_2_ 4.5 mg, vit. B_6_ 3 mg, vit. B_12_ 0.015 mg, biotin 0.1 mg, folic acid 0.8 mg, nicotinic acid 20 mg, pantothenic acid 12 mg, choline 300 mg. HM—*Hermetia illucens* larvae meal; HL—*Hermetia illucens* whole dried larvae.

**Table 2 animals-15-03215-t002:** Chemical analysis and nutritive value of reared pheasant diet with insect larvae meal or whole dried larvae.

Item	Reared (0–4 Weeks)	Grower (58 Weeks) Period					Finisher (9–16 Weeks) Period				
		Control	50 HM	100 HM	50 HL	100 HL	Control	50 HM	100 HM	50 HL	100 HL
Dry matter	896.9	895.4	895.9	896.5	895.8	896.4	894.8	895.2	896.1	895.1	896.3
Crude protein	278.7	230.7	229.3	227.8	228.5	226.6	189.2	187.3	185.5	187.2	186.1
Crude ash	70.1	69.2	69.3	69.4	69.1	69.5	69.2	69.5	69.9	69.4	69.8
Calcium	26.1	26.3	26.5	26.9	26.6	26.8	26.4	26.5	26.8	26.6	26.9
Total phosphorus	7.92	7.81	7.78	7.73	7.77	7.71	7.73	7.71	7.68	7.71	7.67
Ether extract	94.2	82.7	91.3	97.8	90.9	98.1	81.4	87.8	97.6	88.4	96.9
AMEn, MJ kg^−1^ *	12.21	12.07	12.16	12.21	12.15	12.22	11.07	11.12	11.19	11.13	11.19
Fatty acids, % of identified FA											
C12:0	0.08	0.09	8.72	15.93	8.59	16.02	0.12	8.96	16.04	8.81	16.12
C14:0	0.35	0.38	1.16	1.89	1.12	1.92	0.31	1.14	1.83	1.11	2.01
C16:0	20.71	18.13	19.21	21.15	19.16	20.89	17.48	18.31	19.47	18.28	19.41
C18:0	4.16	4.45	4.12	3.97	4.09	3.88	4.58	4.23	4.03	4.21	4.12
C18:1, n-9	36.12	36.79	29.25	22.11	30.11	21.89	35.43	26.03	18.36	25.89	17.76
C18:2, n-6	31.83	33.43	31.82	30.11	31.78	30.27	36.11	35.84	34.73	35.97	35.12
C18:3, n-3	2.68	2.59	2.39	2.26	2.34	2.28	2.62	2.48	2.35	2.45	2.36

* AMEn—metabolizable energy at zero nitrogen balance was calculated by the Fisher and McNab (1987) [49] equations. HM—*Hermetia illucens* larvae meal; HL—*Hermetia illucens* whole dried larvae.

**Table 3 animals-15-03215-t003:** Body weight, average daily gains and feed intake of growing pheasants fed a diet with insect larvae meal or whole dried larvae.

Item	Feeding Groups (F)					Gender (G)		*p* Value		
	Control (*n* = 10)	HM 50 (*n* = 10)	HM 100 (*n* = 10)	HL 50 (*n* = 10)	HL 100 (*n* = 10)	Female (*n* = 10)	Male (*n* = 10)	F	G	F × G
Body weight, g										
-at start (after 4 weeks)	189	197	194	191	191	184 ^b^	208 ^a^	0.431	0.042	ns
-8 weeks	526	551	548	542	536	505 ^b^	584 ^a^	0.207	0.025	0.325
-12 weeks	856	892	885	878	872	794 ^b^	968 ^a^	0.192	0.019	0.386
-16 weeks	1091	1136	1121	1142	1116	1002 ^b^	1254 ^a^	0.196	0.016	0.412
ADG, g										
5–8 weeks	11.7	12.7	12.5	12.3	12.1	11.1	12.9	0.204	0.044	0.386
9–12 weeks	11.5	12.6	12.2	11.9	11.7	10.2	12.8	0.201	0.026	0.389
13–16 weeks	8.6	9.2	8.8	8.9	8.9	7.10	9.80	0.192	0.018	0.405
5–16 weeks	10.5	11.4	11.1	10.7	10.6	9.5	11.8	0.194	0.027	0.378
Feed intake, g d^−1^										
5–8 weeks	40.8	40.1	40.4	40.6	40.7	39.0	41.3	0.406	0.207	0.528
9–12 weeks	59.2	58.4	58.9	58.6	58.8	55.2	60.8	0.462	0.186	0.497
13–16 weeks	73.2	72.1	72.6	72.4	72.6	69.8	73.7	0.423	0.193	0.396
5–16 weeks	57.9	56.1	57.2	56.6	56.8	54.7	58.6	0.405	0.105	0.415

HM—*Hermetia illucens* larvae meal; HL—*Hermetia illucens* whole dried larvae; ^a^, ^b^—differences statistically significant at *p* < 0.05. ADG—average daily gains.

**Table 4 animals-15-03215-t004:** Post-mortem measurements of pheasants fed a diet with insect larvae meal or whole dried larvae.

Item	Feeding Groups (F)					Gender (G)		*p* Value		
	Control (*n* = 10)	50 HM (*n* = 10)	100 HM (*n* = 10)	50 HL (*n* = 10)	100 HL (*n* = 10)	Female (*n* = 10)	Male (*n* = 10)	F	G	F × G
Thigh muscle mass, g	109.4	113.3	114.8	113.9	114.1	98.0 ^b^	128.2 ^a^	0.264	<0.001	0.437
Shank muscle mass, g	100.4	102.2	106.1	105.5	105.9	89.3 ^b^	118.7 ^a^	0.228	<0.001	0.392
Breast muscle mass, g	216.2	210.7	212.2	212.0	211.9	176.7 ^b^	248.5 ^a^	0.381	<0.001	0.412
Head mass, g	52.5	53.2	53.5	53.3	52.7	45.6 ^b^	60.5 ^a^	0.364	<0.001	0.487
Heart mass, g	5.41	5.42	5.41	5.32	5.33	4.12 ^b^	6.61 ^a^	0.433	<0.001	0.645
Liver mass, g	24.5	24.3	24.9	24.8	24.7	23.6 ^b^	25.7 ^a^	0.618	0.031	0.463
Stomach (muscular) mass, g	18.6	18.7	19.2	19.2	19.1	16.6 ^b^	21.3 ^a^	0.285	0.006	0.609

HM—*Hermetia illucens* larvae meal; HL—*Hermetia illucens* whole dried larvae; ^a^, ^b^—differences statistically significant at *p* < 0.05.

**Table 5 animals-15-03215-t005:** Serum lipid profile and cholesterol content in breast and thigh muscles of pheasants fed a diet with insect larvae meal or whole dried larvae.

Item	Feeding Groups (F)					Gender (G)		*p* Value		
	Control (*n* = 10)	50 HM (*n* = 10)	100 HM (*n* = 10)	50 HL (*n* = 10)	100 HL (*n* = 10)	Female (*n* = 10)	Male (*n* = 10)	F	G	F × G
Cholesterol in muscles, mg/100 g:										
Breast	38.4	39.2	39.4	38.9	39.5	39.0	39.2	0.366	0.843	0.682
Thigh	47.9	48.3	48.8	48.7	49.1	48.5	48.6	0.289	0.875	0.712
Serum lipid profile, mmol/L:										
Total cholesterol	4.12	4.19	4.25	4.16	4.21	4.11	4.27	0.203	0.412	0.612
LDL	1.83	1.86	1.88	1.84	1.80	1.85	1.83	0.258	0.568	0.597
HDL	1.87	1.92	1.96	1.91	2.01	1.87	1.99	0.185	0.184	0.465
TG	0.92	0.90	0.89	0.91	0.88	0.91	0.89	0.196	0.418	0.458

HM—*Hermetia illucens* larvae meal; HL—*Hermetia illucens* whole dried larvae; LDL—low density lipoprotein; HDL—high density lipoprotein; TG—triacylglycerol.

**Table 6 animals-15-03215-t006:** Fatty acid composition (% of total FA in breast muscles of pheasants fed a diet with insect larvae meal or whole dried larvae.

Fatty Acids, %	Feeding Groups (F)					Gender (G)		*p* Value		
	Control (*n* = 10)	50 HM (*n* = 10)	100 HM (*n* = 10)	50 HL (*n* = 10)	100 HL (*n* = 10)	Female (*n* = 10)	Male (*n* = 10)	F	G	F × G
C 12:0	0.17	0.45	0.53	0.44	0.58	0.38	0.48	0.001	0.057	0.431
C 14:0	1.32	1.23	1.14	1.24	1.15	1.18	1.26	0.072	0.125	0.387
C 14:1	0.07	0.06	0.05	0.05	0.05	0.07	0.05	0.232	0.152	0.298
C 16:0	21.63	21.05	22.04	21.78	22.11	21.51	21.93	0.327	0.423	0.396
C 16:1	3.21	3.14	3.11	3.28	3.31	3.15	3.27	0.278	0.315	0.458
C 17:1, n-7	0.12	0.11	0.10	0.09	0.08	0.11	0.09	0.105	0.268	0.612
C 18:0	14.56	13.79	13.94	13.96	14.07	13.87	14.25	0.183	0.217	0.326
C 18:1, n-9	24.34	23.91	22.17	23.45	22.86	22.75	23.95	0.175	0.142	0.094
C 18:1, n-7	2.42	2.41	2.53	2.38	2.29	2.46	2.36	0.347	0.268	0.543
C 18-2, n-6	20.45	20.93	21.04	21.02	20.73	22.32 ^a^	19.34 ^b^	0.412	0.045	0.121
C 18:3, n-3	0.95 ^b^	1.21 ^ab^	1.34 ^ab^	1.28 ^ab^	1.54 ^a^	1.14 ^b^	1.38 ^a^	0.042	0.039	0.158
C 20:0	0.39 ^a^	0.29 ^ab^	0.21 ^b^	0.28 ^ab^	0.24 ^b^	0.27	0.29	0.033	0.312	0.397
C 20:1, n-9	0.23	0.17	0.16	0.21	0.23	0.22	0.18	0.137	0.195	0.412
C 20:2, n-6	0.15	0.14	0.12	0.14	0.15	0.16	0.12	0.379	0.095	0.312
C 20:4, n-6	7.24	7.11	7.18	7.21	7.19	7.29	7.09	0.335	0.286	0.438
C 20:5, n-3	0.21	0.23	0.25	0.24	0.26	0.25	0.23	0.412	0.524	0.735
C 22:5, n-3	0.11	0.12	0.13	0.14	0.15	0.15	0.11	0.453	0.424	0.785
C 22:6, n-3	0.36	0.39	0.41	0.40	0.44	0.41	0.39	0.346	0.512	0.658
SFA	38.07	36.81	37.86	37.70	38.15	37.21	38.21	0.278	0.296	0.548
MUFA	30.39	29.80	28.12	29.46	28.82	28.76	29.90	0.292	0.345	0.128
PUFA	29.47	30.13	30.47	30.43	30.46	31.72	28.66	0.326	0.126	0.387
PUFA, n-3	1.63 ^c^	1.95 ^b^	2.13 ^ab^	2.06 ^b^	2.39 ^a^	1.95	2.11	0.032	0.189	0.612
PUFA, n-6	27.84	28.18	28.34	28.37	28.07	29.77 ^a^	26.55 ^b^	0.147	0.046	0.386
PUFA, n-6/n-3	17.08 ^a^	14.45 ^b^	13.31 ^b^	13.77 ^b^	11.74 c	15.27 ^a^	12.58 ^b^	0.025	0.042	0.279

HM—*Hermetia illucens* larvae meal; HL—*Hermetia illucens* whole dried larvae; ^a^, ^b^, ^c^—differences statistically significant at *p* < 0.05.

**Table 7 animals-15-03215-t007:** Fatty acid composition (% of total FA) in thigh muscles of pheasants fed with insect meal or larvae.

Fatty Acids, %	Feeding Groups (F)					Gender (G)		*p* Value		
	Control (*n* = 10)	50 HM (*n* = 10)	100 HM (*n* = 10)	50 HL (*n* = 10)	100 HL (*n* = 10)	Female (*n* = 10)	Male (*n* = 10)	F	G	F × G
C 12:0	0.15 ^b^	0.36 ^a^	0.42 ^a^	0.33 ^a^	0.45 ^a^	0.36	0.32	0.001	0.098	0.426
C 14:0	1.64	1.74	1.82	1.75	1.91	1.72	1.82	0.063	0.119	0.437
C 14:1	0.12	0.13	0.15	0.14	0.16	0.15	0.13	0.226	0.327	0.392
C 16:0	27.11	27.58	27.67	27.55	27.84	27.05	28.05	0.384	0.223	0.412
C 16:1	2.57	2.43	2.26	2.42	2.31	2.47	2.33	0.168	0.345	0.487
C 17:1, n-7	0.15	0.14	0.12	0.14	0.13	0.15	0.13	0.135	0.326	0.645
C 18:0	18.24	17.67	18.35	18.36	18.71	18.07	18.47	0.208	0.314	0.463
C 18:1, n-9	31.23	30.84	30.11	31.02	29.97	31.03	30.13	0.182	0.164	0.109
C 18:1, n-7	1.89	1.78	1.73	1.68	1.72	1.79	1.73	0.369	0.462	0.654
C 18-2, n-6	10.45	10.93	11.04	11.02	10.73	11.38	10.33	0.396	0.065	0.226
C 18:3, n-3	0.64 ^b^	0.73 ^b^	0.87 ^ab^	0.78 ^b^	0.96 ^a^	0.84	0.76	0.041	0.109	0.352
C 20:0	0.12	0.14	0.15	0.13	0.15	0.11	0.17	0.295	0.112	0.412
C 20:1, n-9	0.19	0.17	0.18	0.15	0.17	0.20	0.14	0.232	0.091	0.436
C 20:2, n-6	0.21	0.22	0.23	0.19	0.22	0.23	0.19	0.385	0.209	0.431
C 20:4, n-6	2.23 ^a^	1.94 ^ab^	1.82 ^ab^	1.87 ^ab^	1.69 ^b^	2.06	1.76	0.034	0.148	0.134
C 20:5, n-3	0.26	0.26	0.25	0.28	0.29	0.28	0.26	0.425	0.584	0.738
C 22:5, n-3	0.15	0.16	0.15	0.18	0.19	0.18	0.16	0.492	0.422	0.756
C 22:6, n-3	0.39 ^a^	0.38 ^a^	0.35 ^a^	0.38 ^a^	0.27 ^b^	0.37	0.33	0.041	0.505	0.651
SFA	47.26	47.49	48.41	48.12	49.06	47.31	48.83	0.124	0.209	0.512
MUFA	36.15	35.49	34.55	35.55	34.46	35.79	34.59	0.272	0.348	0.232
PUFA	14.33	14.62	14.71	14.7	14.35	15.34	13.79	0.463	0.122	0.379
PUFA, n-3	1.44	1.53	1.62	1.62	1.71	1.67	1.51	0.124	0.201	0.513
PUFA, n-6	12.89	13.09	13.09	13.08	12.64	13.67	12.28	0.195	0.094	0.506
PUFA, n-6/n-3	8.95 ^a^	8.56 ^ab^	8.08 ^b^	8.07 ^b^	7.39 ^b^	8.19	8.13	0.034	0.448	0.307

HM—*Hermetia illucens* larvae meal; HL—*Hermetia illucens* whole dried larvae; ^a^, ^b^—differences statistically significant at *p* < 0.05.

## Data Availability

The datasets generated during and/or analysed during the current study are available from the corresponding author upon reasonable request.

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
