# Peer review of "Effect of Meal and Whole Larvae of Black Soldier Fly (*Hermetia illucens*) on the Performance, Blood Lipid Profile, Slaughter Characteristics, Sensory Properties and Fatty Acid Composition of Pheasant (*Phasianus colchicus* L.) Muscles"

_animals, 2025, doi:10.3390/ani15213215_

Round 1
Reviewer 1 Report
Comments and Suggestions for Authors
Comments and suggestions are presented in the attached document.

Author Response
Response to the reviewer's comments

Reviewer 2 Report
Comments and Suggestions for Authors
Dear authors,
here is a peer review of the manuscript submitted to the journal Animals, entitled
“Effect of meal and whole larvae of black soldier fly (Hermetia illucens) on performance, blood lipid profile, slaughter, sensory properties and fatty acid composition of pheasant (Phasianus colchicus L.) muscles”
The manuscript, which is a type of scientific article, contains 6 tables and 4 figures on 21 pages of text, the authors cite 74 references.
Overall view of the article
The manuscript deals with the current and scientifically very relevant issue of the use of insect protein (Hermetia illucens) in pheasant feed. The topic fits into the current trend of finding sustainable sources of protein for animal production and at the same time brings new knowledge about the nutritional quality of meat from game kept in captivity. The study is methodologically well structured, clearly defines the goal and logically develops the research hypothesis on the possible replacement of soybean meal with insect meal or whole larvae.
From this perspective, it is an interesting topic with citation potential.
The work brings original results, especially for the species Phasianus colchicus, which is less studied compared to commercial poultry. The contribution is also a comprehensive evaluation — from growth performance to blood lipid profile to sensory properties of meat and fatty acid composition.
Methodology
The description of the methodology is detailed and transparent. The authors correctly state:
the ethical approval of the experiment (Lublin, No. 8/2024),
the precise characteristics of the experimental design (5 groups: control, 50% and 100% replacement of soybean meal or larvae),
analytical procedures according to AOAC and methodology for determining the lipid profile, including GC-FID conditions.
The sensory evaluation methodology according to Baryłko-Piekielna is suitable, but the five-member panel is relatively small - I recommend stating the qualifications of the evaluators (e.g. training according to ISO 8586) and the method of repetition. It would also be appropriate to mention the method of homogenization of samples and calibration of the chromatography device.
Sensory analysis is the soft spot of this study.
Statistical processing using ANOVA and Tukey's test is standard, however, information on the number of animals in individual groups is missing, which makes it difficult to assess statistical power. I recommend adding.
Results and discussion
The presentation of the results is clear, the tables are logically arranged. The results show that:
replacement of soybean meal with 50% and 100% BSF products did not negatively affect growth parameters or feed consumption;
no significant differences were found in organ weights or blood lipids;
the content of n-3 PUFA in breast muscle slightly increased, while the n-6/n-3 ratio improved;
the sensory properties remained preserved.
The interpretation is correct, well placed in the literary context. The discussion appropriately compares the results with previous studies in other species (chickens, quail, partridge). However, it would be useful to analyze in more detail the nutritional significance of the increased content of lauric acid (C12:0), which has antimicrobial effects, but in humans can be evaluated ambivalently in terms of LDL-cholesterol.
Language and style
The text is linguistically cultivated and scientifically correct. Occasionally there are minor grammatical inaccuracies (e.g. "were not significantly dependent on the substitution of soybean meal with BSF products" could be formulated more concisely), but these shortcomings do not affect the understandability. I recommend a stylistic revision by native English speakers to increase the linguistic level of the manuscript.
Recommendations
- Number of animals - it is necessary to state N for each group.
Replication of sensory analysis - add the number of repetitions and qualification of the panel. - Fatty acids – expand the interpretation of the health implications of increased C12:0 and C14:0.
- There is no graphical summary (e.g. PCA or heatmap of fatty acids) that would facilitate the reader's orientation.
- Conclusion – it is appropriate to add a quantitative statement ("BSF meal can replace up to 100% of soybean meal without significant changes in performance").
- I recommend minor changes to the citations (e.g. the format of DOI and references to previous works by the same authors should be unified).
- However, there is a problem with the style and formatting of the article in the usual MDPI style.
- Also, the images are meaninglessly labeled Figure 1a - 1d, why? Why are they not Figures 1 to 4?
- Their size and graphic design are inappropriate. I recommend looking at other articles in MDPI Animals.
- The tables are not clear in relation to the manuscript and need to be adjusted to the page.
- I recommend checking the text as a whole, there are errors, e.g. in the tables and their titles (Line 223: units, the mathematical sign times is missing "Table 1. Ingredients (g kg-1)..."
- Line 152: "A 5-point scale was used to assess the aroma and intensity, with the following points assigned for aroma: 5 - very distinct, 4 - distinct, 3 - slightly distinct, 2 - perceptible, 1 - imperceptible. In turn, when assessing the succulence, the point ranges indicated the following: 5 - succulent, 4 - moderately succulent, 3 - slightly succulent, 2 - slightly dry, 1 - dry. The tenderness scale included the following scores: 5 - very tender, 4 - tender, 3 - slightly tender, 2 - firm, and 1 - very firm. "
- I see the most fundamental mistake in the shallow sensory analysis, although the methodology is cited, I do not think that the 5-point scale is ideal. However, I respect the methodology used, if it were described correctly. The graphs show smell and aroma. What is the difference between them? The methodology only shows aroma and there is no mention of smell...
Final opinion
The manuscript represents a scientifically beneficial and experimentally solid study on the use of BSF products in pheasant feeds. The results provide important insights for the development of entomoproteins in practice and for increasing the sustainability of small game farming.
After major formal and stylistic adjustments, I recommend accepting the manuscript for publication.
Author Response
Response to the reviewer's comments

Round 2
Reviewer 1 Report
Comments and Suggestions for Authors
Suggestions and comments are given in the attached file.

Author Response
In response to the reviewers' comments, the English language has been improved, and I have submitted a native speaker certificate. Tables and figures have also been corrected in accordance with the reviewers' suggestions.
Reviewer 2 Report
Comments and Suggestions for Authors
I have no comments now, the authors have reworked the problematic passages and improved the manuscript graphically and formally.
Author Response

(The authors gave the same response as above.)
